# Fostering active living and healthy eating through understanding physical activity and dietary behaviours of Arabic-speaking adults: a cross-sectional study from the Middle East

Tam Truong Donnelly,[1] Tak Shing Fung,[2] Al-Anoud bint Mohammad Al-Thani[3]

[1]Faculty of Nursing and Medicine, University of Calgary, Calgary, Alberta, Canada
[2]Research Computing Services, University of Calgary, Calgary, Alberta, Canada
[3]Ministry of Public Health, Health Promotion and Non Communication Diseases, Doha, Qatar

**Correspondence to**
Dr Tam Truong Donnelly;
tdonnell@ucalgary.ca

## ABSTRACT

**Objectives** Physical inactivity and unhealthy diets increase the risk for diabetes, cardiovascular diseases and cancer. Many people in Qatar are sedentary and consume diets high in fats, salt and sugar. The purpose of this study was to determine physical activity levels, food habits and understand the variables that might predict physical activity and healthy eating behaviours among Arabic-speaking adults living in the State of Qatar.

**Setting** A cross-sectional community-based survey was conducted with 1606 Arab adults ≥18 years of age from March 2013 to June 2015. Using a non-probability sampling technique, participants were recruited from three universities and five primary healthcare centres in Qatar. Participants were interviewed using a structured survey questionnaire. The survey included questions regarding demography, clinical characteristics and the participant's daily dietary practice. Physical activity level was assessed by the Arabic version of the International Physical Activity Questionnaire. Statistical analysis was performed using SPSS V.22.0.

**Results** Of 1606 participants, 50.1% were men and 49.9% were women. The participants' mean (SD) body mass index was 28.03 (5.85) Kg/m². Two-thirds of the participants were either overweight (36.4%) or obese (33.6%). Within the 7 days prior to completing the questionnaire, 64% and 39.9% of study participants did not engage in vigorous or moderate physical activity, respectively. Within the 7 days prior to completing the questionnaire, the mean (SD) time for vigorous physical activity was 31.12 (59.28) min, 46.87 (63.01) min for moderate physical activity, and 42.01 (47.04) min for walking. One-third of the participants consumed fresh fruits and vegetables once or more daily, and fish, beef or chicken 2–4 times weekly. One quarter of the participants ate pasta, cakes or pastries 2–4 times weekly, and 40.6% of them ate white bread daily.

**Conclusions** Participants exhibited insufficient physical activity and poor dietary habits. There is a need for a nationwide health promotion programme to promote a healthier lifestyle. The information from this study can inform public health policies, programmes and services in Qatar and other Middle Eastern countries.

### Strengths and limitations of this study

► Large sample size (n=1606) that is representative of the Arab Qatari population and face-to-face interviews with all participants are major strengths of this study.
► Non-probability convenience sampling limits the generalisability of the study findings.
► This study only reports data pertaining to Arabic-speaking adults. Younger participants (age <18 years) might give different perspectives on their physical activity and dietary behaviour.

## INTRODUCTION

Physical inactivity and unhealthy diets are major health and economic problems worldwide and are important modifiable risk factors for non-communicable diseases such as type 2 diabetes, cardiovascular disease and some cancers.[1–3] In a recent study, nine modifiable risk factors that accounted for more than 90% of the risk for acute myocardial infarction were identified as: sedentary lifestyle, history of hypertension, diabetes, smoking, abdominal obesity, occasional/non-daily consumption of fruits and vegetables, psychosocial factors, regular alcohol consumption and a raised apolipoprotein B/ApoA1 ratio.[4] These risk factors were found to be associated with myocardial infarction irrespective of gender and age. Similarly, in the Arabian Gulf countries, the most eminent risk factors of non-communicable diseases were known as inadequate intake of fruit and vegetables, being overweight or obese, physical inactivity, high blood pressure, high blood cholesterol and tobacco use.[5] In 2013, the cost of physical inactivity for healthcare systems was $53.8 billion worldwide.[2] In Qatar, the annual direct and indirect costs attributable to physical inactivity were estimated as $60.7 million

in 2013.[1] Furthermore, it costs people's lives, well-being, quality of life and health of their families and caregivers.

According to the WHO's[6] guidelines for physical activity (PA), healthy adults aged 18–64 years should accumulate at least 150 min of moderate-intensity aerobic activity per week or 75 min of vigorous-intensity activity or an equivalent combination of moderate-intensity and vigorous-intensity aerobic activity in bouts of 10 min or more. Similarly, the State of Qatar National Physical Activity Guidelines recommends that healthy adults aged 18–64 years should do 5 days or more of 30–60 min of moderate exercise per week or 3 days or more of 20–60 min of vigorous exercise per week.[7] However, the results of a study in the Gulf Cooperation Council (GCC) countries showed that only 39%–42.1% of men and 26.3%–28.4% of women were physically active for at least 150 min per week.[8] According to the Qatar STEPwise survey conducted in 2012, 45.9% of the participants were engaged in low levels of activity per week, and 31.3% were engaged in high levels of activity per week. The median time spent in PA on an average day was 37 min. Of all the study participants, 71.3% were not engaged in vigorous activity.[9] According to WHO, the highest prevalence of insufficient PA (31%) was located in the Eastern Mediterranean Region. In Qatar, the prevalence of insufficient PA in adults was reported as 41.6%.[10] Ramirez et al[11] observed that one in five patients with the highest cardiac risk did not perceive the need to improve their physical health.

According to a recent systematic review, regular intake of fruits and vegetables was inversely associated with high systolic blood pressure, abdominal obesity, triglycerides, high-density lipoprotein cholesterol and metabolic syndrome.[12] Along with the WHO healthy diet (HD) guidelines, Qatar's Food-based Dietary Guidelines recommend that people should eat healthy choices from different food groups, limit sugar, salt and fat, and maintain a healthy weight.[13] However, urbanisation alongside the modernisation and westernisation in Arab countries have shaped dietary habits; people in Qatar have ready access to foods high in sugar, fats and salts (eg, fast food and carbonated beverages).[14–16] According to the Qatar STEPwise approach to chronic disease risk factor surveillance, adult participants aged 18–64 years consumed 0.8 servings of fruit and 1.4 servings of vegetables on average per day, and 91% ate less than five servings of fruit and/or vegetables on average per day.[9] Situated on the Gulf Coast of the Arabian Peninsula (figure 1), with a population of nearly 2.6 million,[17] the State of Qatar had the fastest growing economy in the Gulf region and the highest gross domestic product per capita due to its abundant oil and natural gas revenues.[18] This steep economic growth over the decades seemed to influence availability of food and patterns of food consumption. However, the current literature on dietary practice of the Arab Qatari population is scarce.

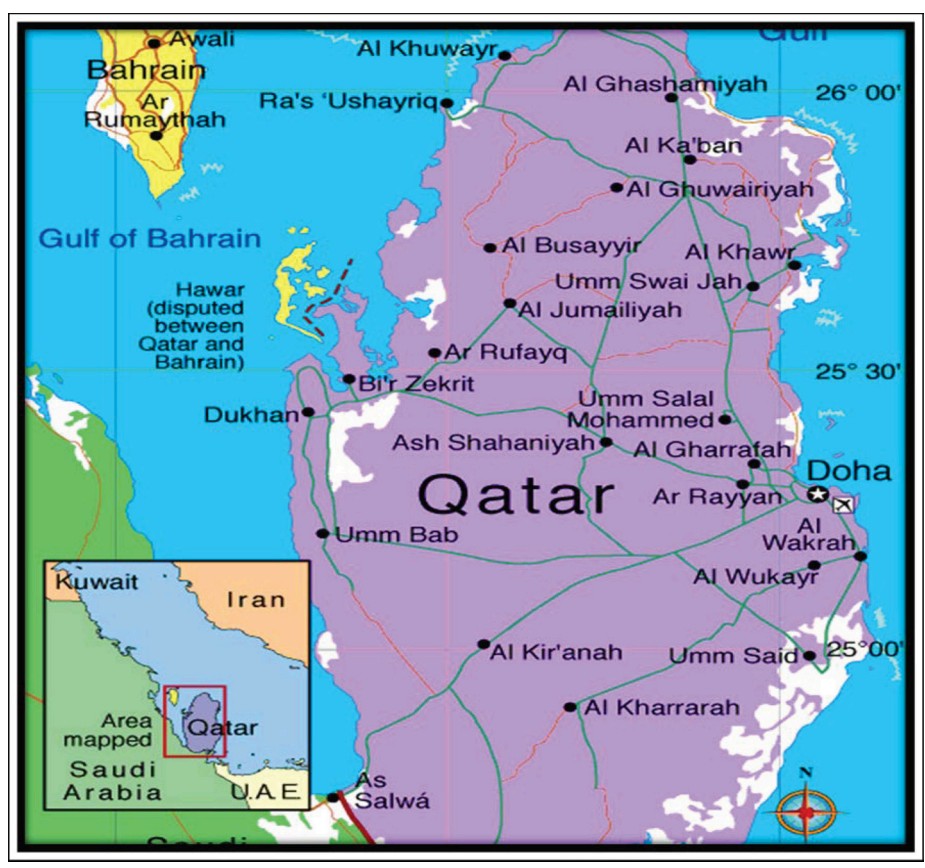

**Figure 1** Gulf Coast of the Arabian Peninsula.

Given that obesity is linked to physical inactivity and/or unhealthy diet and that the fundamental cause of obesity and overweight is an energy imbalance between calories consumed and calories burned,[19] lifestyle changes such as increased PA, decreased sedentary lifestyle and healthy dietary changes are necessary for weight management.[20] According to WHO,[19] more than 1.9 billion adults worldwide were overweight in 2014, of those, over 600 million adults were obese. Currently, there is limited information on the level of PA and dietary habits among adults in Qatar. The main purpose of this study was to determine the current PA levels and dietary habits, and understand the variables that might predict PA and healthy eating behaviours among Arabic-speaking adults living in the State of Qatar.

## METHODS
### Participants

A cross-sectional, community-based survey with Arabic-speaking adults living in Qatar was conducted from March 2013 to June 2015. Participants were eligible for inclusion in the study if they were (1) 18 years or older, (2) self-identified as an Arabic speaker, (3) born and/or raised in Qatar, or lived in Qatar for at least 5 years and (4) willing to commit a minimum of 60 min for the questionnaire. We approached and recruited participants at the designated data-collection sites—three universities and five healthcare centres in the capital Doha and areas South and North of Doha in Qatar, to ensure diversity of participants that can closely represent the general Arab population in Qatar. Although random selection helps to reduce selection biases, this sampling technique was not feasible for this research. It is difficult to gain access to Arab populations because of sociocultural beliefs and practices that value privacy. Hence, we used a purposeful, non-probability convenient sampling technique. We realised that this increased the risk of selection biases and limited the generalisability of the research findings. To help offset these limitations, we recruited and randomly selected participants at different times of the days, weeks or months of the year at the designated data-collection sites.

Trained interviewers who were fluent in both Arabic and English identified eligible participants based on the inclusion criteria. Providing the eligible individual answered 'yes' to the screening questions and wished to continue, the interviewer: (1) provided the participant with a short explanation of the study, (2) advised the participant that his/her participation was strictly voluntary, (3) advised him/her that measures would be taken to help ensure confidentiality and (4) answered any of the participant's questions. When the participant agreed to participate, this was considered consent by assent. The interviewer then enrolled the participant in the study and administered the survey. Recruitment continued until the determined sample size was reached.

According to the Qatar census (2010) data, there were 1 008 937 women and men (843 441 men and 165 496 women) between the ages of 15 and 75+ years living in Doha, Al Wakrah (South of Qatar), Al Khor and Al Thakhira (North of Qatar). Based on the Cochran's formula using a margin of error of 3.5% (95% CI), a sample size of 781 women and 784 men was determined to be a representative sample of the Arabic-speaking adults living in the above four regions.[21] Please note that we did not have access to census data that included individuals aged 18 and older. The census data we used includes ages of 15+ years.

We acknowledged that participants might not be familiar with the research process because little research has been conducted in Qatar, thus, the research team members carefully explained the project to participants. To enhance accessibility, the project information was available in both Arabic and English. Prior to the start of the study, an introductory letter was sent to the university and community health clinic sites. Formal and informal presentations were provided to all staff at each site at the beginning of the project. Written informed consent was waived, but verbal consent to participate in the study was obtained from each participant. All participants were assured that information would be confidential. No incentive was given to the participants.

### Data collection

Data collection was carried out using a structured survey questionnaire designed for this study. The survey instrument was developed based on literature reviewed. The survey questionnaire had four sections. The first section consisted of questions asking for demographic information of the participants. The second section contained questions regarding levels of PA using the International Physical Activity questionnaire (IPAQ).[22] The third section pertained to the participant's current and previous history of chronic diseases. The last section consisted of questions that could assess the participant's daily food intake.

All of the study material and questions were translated to Arabic and back translated to English to ensure lexical equivalence. To ensure the questionnaire items were relevant to the context in Qatar, we conducted a pilot study using six focus groups stratified by age (18–30 years, 31–50 years, 51+ years) and sex (male, female). We conducted separate focus groups for men and women to be culturally appropriate and stratified by age to group people who were likely to have similar life experiences. Following the focus groups, the questionnaire was further pilot tested with 12 men and 12 women recruited from urban and semiurban settings who represented different age groups. Based on the pilot test, we further refined the data-collection protocol and the questionnaire. Experienced and specially trained bilingual (Arabic and English) research assistants conducted structured interviews in Arabic.

## Body mass index

Obesity status was determined by the body mass index (BMI). BMI represents the standard method used by both the WHO and the governmental health sectors of the majority of nations to determine whether a person is obese or overweight. BMI is determined by dividing the individual's weight measured in kilograms by his/her height measured in metres squared. Obesity and overweight were classified based on BMI as: underweight ≤18.5 kg/m², normal weight 18.5–24.9 kg/m², overweight 25–29.9 kg/m², obese class I 30–34.9 kg/m², obese class II 35–39.9 kg/m², obese class III ≥40 kg/m².[23]

## Physical activity

IPAQ asks participants to recall their physical activities in the last 7 days. The results allow researchers to classify participants into one of the three levels of PA (ie, low, moderate or high). IPAQ has evidence for reliability and validity for monitoring levels of PA among adults aged 18–65 years in diverse settings.[22 23] For this study, we used the short telephone version which was translated into Arabic. The IPAQ website provides protocols related to data cleaning, processing, scoring and translation.[23]

## Dietary behaviour

Dietary behaviour was explored based on data collected using the study questionnaire which provides details about daily or weekly frequency of consumption for various food groups including fruits, vegetables, proteins (fish, beef, lamb or chicken), carbohydrates (bread, cereals or pasta), milk products and other food items (soft drinks, juices or nuts) consumption, including number of servings per day. For example, questions were asked: How often do you eat fresh fruits/green leafy vegetables/fish/cakes and pastries, etc? Participants could respond: never, seldom, once a week, 2–4 times a week, 5–6 times a week, once or more daily, don't know.

Health status was reviewed based on data collected using the study questionnaire. The questionnaire included questions regarding history of chronic diseases such as hypertension, diabetes, dyslipidaemia, stroke, obesity, history of fractures and allergies.

## Data analysis

Data analyses were conducted by a senior biostatistician using SPSS (IBM SPSS Statistics V.22.0). Descriptive statistical analyses (frequencies, means and SD for interval variables, frequency and proportion for categorical variables) were performed for the study variables. Data were expressed in frequencies for questionnaire responses calculated for all variables in numbers and percentages. The Mann-Whitney test was used to compare differences between two groups with ordinal data, $X^2$ test was used to compare difference between two groups with categorical data and an independent sample t-test was used to compare group differences with interval data. Multivariate logistic regression analysis was performed to identify the variables that predicted PA, HD engagement and IPAQ group. Independent variables, such as living area, marital status, nationality, age group, education, health status, household income and sex were selected using the criteria for the method of forward stepwise (Wald $\chi^2$, $p_{in}$=0.05, $p_{out}$=0.10). The equation used to build the model is: $\ln(p/(1-p)) = \alpha_0 + \alpha_1 x_1 + \alpha_2 x_2 + \ldots \ldots \alpha_k x_k$, where p=probability of PA, HD engagement and IPAQ group, respectively, $x_1, x_2, \ldots \ldots x_k$ are significant predictors after forward stepwise logistic regression procedure. Statistical significance levels were established at alpha=0.05.

## RESULTS

### Participants characteristics

We approached 3081 participants, of which 1606 participants who met the study's inclusion criteria participated in the study (response rate of 52.1%). Eight hundred and four (50.1%) were men and 802 (49.9%) were women. The majority of study participants were under the age of 60 years. Approximately 20.8% of the participants belonged to the age group of 40–49 years. Almost all participants were Muslim (99.1%). More than half of the study participants were married (59.7%), and 40.3% were without a spouse. A total of 30.3% of the study's participants were Qatari nationals. Close to half of the participants (43.8%) came from North African countries (Egypt, Libya, Tunisia, Algeria, Morocco and Mauritania). The citizens of the Levant countries (Syria, Lebanon, Palestine and Jordan) constituted 19.9%. The rest of the participants were from the GCC and other countries. Overall, more than one-third of the participants completed university (37.8%) and 42% completed high school. Nearly one-third of the participants' (32.7%) household income ranged between US$2746 and US$5491 (QAR 10 000–19 999) per month. Most participants were non-smokers (79.1%), 77.9% of the participants resided in Doha, 9.4% lived in Al Wakrah and 12.6% were residents of Al Khor (table 1).

Four MET-min/week (walking, moderate, vigorous and total) were calculated for each participant. Based on the IPAQ criteria,[23] each participant was classified into one of the three PA categories: low, moderate and high. IPAQ group: low (n=348, 21.7%), moderate (n=854, 53.2%), high (n=390, 24.3%). Fourteen participants (0.9%) could not be classified into any of the IPAQ groups due to missing values. The mean (SD) of walking, moderate, vigorous and total MET-min/week were 562.12 (791.31), 745.03 (1330.51), 809.60 (1803.51) and 2062.80 (2617.40) respectively (table 2).

### Participants' BMI

The study's research assistant measured each participant's height and weight and calculated the participant's BMI according to the guideline. The mean (SD) BMI of the study participants was 28.03 (5.85) kg/m² and ranged from 12.8 kg/m² to 65.44 kg/m². Overall prevalence rates of underweight, normal weight and overweight were 44 (2.8%), 433 (27.2%), and 580 (36.4%), respectively.

**Table 1** Selected demographic characteristics of participants (n=1606)

| Characteristics | Total, n (%) | Male, n (%) | Female, n (%) | P values |
|---|---|---|---|---|
| **Age (years)*** | | | | |
| 18–29 | 517 (32.3) | 261 (32.6) | 256 (32) | |
| 30–39 | 557 (34.8) | 258 (32.3) | 299 (37.4) | |
| 40–49 | 333 (20.8) | 145 (18.1) | 188 (23.5) | |
| 50–59 | 136 (8.5) | 90 (11.3) | 46 (5.8) | |
| 60–69 | 49 (3.1) | 39 (4.9) | 10 (1.3) | |
| 70+ | 8 (0.5) | 7 (0.9) | 1 (0.1) | <0.001† |
| **Marital status** | | | | |
| Single/never married | 595 (37) | 307 (38.2) | 288 (35.9) | |
| Married | 958 (59.7) | 487 (60.6) | 471 (58.7) | <0.001† |
| Separated/divorced/widowed | 53 (3.3) | 10 (1.2) | 43 (5.4) | |
| **Nationality** | | | | |
| Qatari | 487 (30.3) | 190 (23.6) | 297 (37.1) | |
| North Africa | 704 (43.8) | 415 (51.7) | 289 (36) | |
| Levant | 320 (19.9) | 159 (19.7) | 161 (20.1) | |
| Other GCC countries | 33 (2) | 12 (1.5) | 21 (2.6) | |
| Other | 61 (3.8) | 28 (3.4) | 33 (4.2) | <0.001† |
| **Level of education‡** | | | | |
| Never went to school | 18 (1.1) | 8 (1) | 10 (1.2) | |
| Primary school | 115 (7.2) | 45 (5.6) | 70 (8.7) | |
| High school | 685 (42) | 330 (41.1) | 355 (44.3) | |
| Trade school | 34 (2.1) | 26 (3.2) | 8 (1) | |
| University | 607 (37.8) | 319 (39.7) | 288 (36) | |
| Other | 145 (9) | 75 (9.3) | 70 (8.7) | 0.003† |
| **Monthly income in US$§** | | | | |
| <274 | 7 (0.6) | 5 (0.8) | 2 (0.3) | |
| 275–2745 | 192 (15.7) | 103 (16.9) | 89 (14.5) | |
| 2746–5491 | 401 (32.7) | 224 (36.7) | 177 (28.8) | |
| 5492–8237 | 292 (23.8) | 140 (23) | 152 (24.7) | |
| 8238–10983 | 133 (10.9) | 57 (9.3) | 76 (12.4) | |
| 10984–13729 | 73 (6) | 26 (4.3) | 47 (7.6) | |
| 13730–16476 | 49 (4) | 22 (3.6) | 27 (4.4) | |
| >16476 | 78 (6.4) | 33 (5.4) | 45 (7.3) | 0.007† |
| **Smoking¶** | | | | |
| Yes | 334 (20.9) | 293 (36.8) | 41 (5.1) | |
| No | 1263 (79.1) | 504 (63.2) | 759 (94.9) | <0.001† |
| **Living area** | | | | |
| Al Doha | 1251 (77.9) | 626 (77.9) | 625 (77.9) | |
| Al Wakrah | 151 (9.4) | 78 (9.7) | 73 (9.1) | |
| Al Khor | 202 (12.6) | 98 (12.2) | 104 (13) | |
| Al Thakhira | 2 (0.1) | 2 (0.2) | 0 (0) | 0.505 |

*Six participants did not answer this question.
†Significant at α=0.05 level.
‡Two participants did not answer this question.
§Three hundred and eighty-one participants did not answer this question.
¶Nine participants did not answer this question.
GCC, Gulf Cooperation Council.

**Table 2** Relationship between selected demographic characteristics of participants, health status and medical history with physical activity category (low, moderate, high)

| Characteristics | Total, n (%) | Low, n (%) | Moderate, n (%) | High, n (%) | P values |
|---|---|---|---|---|---|
| **Age (years)** | | | | | |
| 18–29 | 507 (32.0) | 95 (27.5) | 218 (25.6) | 194 (49.7) | |
| 30–39 | 556 (35.1) | 120 (34.8) | 327 (38.4) | 109 (27.9) | |
| 40–49 | 330 (20.8) | 78 (22.6) | 201 (23.6) | 51 (13.1) | |
| 50–59 | 136 (8.6) | 40 (11.6) | 70 (8.2) | 26 (6.7) | |
| 60–69 | 49 (3.1) | 10 (2.9) | 29 (3.4) | 10 (2.6) | |
| 70+ | 8 (0.5) | 2 (0.6) | 6 (0.7) | 0 (0.0) | <0.001* |
| **Marital status** | | | | | |
| Single/never married | 586 (36.8) | 117 (33.6) | 274 (32.1) | 195 (50.0) | |
| Married | 955 (60.0) | 216 (62.1) | 551 (64.5) | 188 (48.2) | |
| Separated/divorced/widowed | 51 (3.2) | 15 (4.3) | 29 (3.4) | 7 (1.8) | <0.001* |
| **Nationality** | | | | | |
| Qatari | 482 (30.3) | 131 (37.6) | 244 (28.6) | 107 (27.4) | |
| North Africa | 698 (43.9) | 131 (37.6) | 388 (45.5) | 179 (45.9) | |
| Levant | 318 (20.0) | 65 (18.7) | 181 (21.2) | 72 (18.5) | |
| Other GCC countries | 33 (2.1) | 7 (2.0) | 14 (1.6) | 12 (3.1) | |
| Other | 60(3.8) | 14 (4.0) | 26 (3.0) | 20 (5.1) | 0.014* |
| **Level of education** | | | | | |
| Never went to school | 18 (1.1) | 3 (0.9) | 9 (1.1) | 6 (1.5) | |
| Primary school | 115 (7.2) | 30 (8.6) | 59 (6.9) | 26 (6.7) | |
| High school | 676 (42.5) | 135 (38.9) | 319 (37.4) | 222 (56.9) | |
| Trade school | 34 (2.1) | 10 (2.9) | 16 (1.9) | 8 (2.1) | |
| University | 602 (37.9) | 140 (40.3) | 371 (43.5) | 91 (23.3) | |
| Other | 145 (9.1) | 29 (8.4) | 79 (9.3) | 37 (9.5) | <0.001* |
| **Monthly income in US$** | | | | | |
| <274 | 7 (0.6) | 3 (1.2) | 3 (0.4) | 1 (0.3) | |
| 275–2745 | 192 (15.7) | 31 (12.3) | 93 (13.8) | 68 (23.2) | |
| 2746–5491 | 397 (32.5) | 81 (32.1) | 221 (32.7) | 95 (32.4) | |
| 5492–8237 | 291 (23.9) | 62 (24.6) | 175 (25.9) | 54 (18.4) | |
| 8238–10 983 | 133 (10.9) | 28 (11.1) | 77 (11.4) | 28 (9.6) | |
| 10 984–13 729 | 73 (6.0) | 25 (9.9) | 37 (5.5) | 11 (3.8) | |
| 13 730–16 476 | 49 (4.0) | 6 (2.4) | 28 (4.1) | 15 (5.1) | |
| >16 476 | 78 (6.4) | 16 (6.3) | 41 (6.1) | 21 (7.2) | 0.002* |
| **Smoking** | | | | | |
| Yes | 332 (21.0) | 67 (19.5) | 183 (21.6) | 82 (21.0) | |
| No | 1251 (79.0) | 277 (80.5) | 666 (78.4) | 308 (79.0) | 0.727 |
| **Living area** | | | | | |
| Al Doha | 1239 (77.8) | 279 (80.2) | 630 (73.8) | 330 (84.6) | |
| Al Wakrah | 149 (9.4) | 30 (8.6) | 98 (11.5) | 21 (5.4) | |
| Al Khor | 202 (12.7) | 38 (10.9) | 126 (14.8) | 38 (9.7) | |
| Al Thakhira | 2 (0.1) | 1 (0.3) | 0 (0) | 1 (0.3) | 0.001* |
| **Health status** | | | | | |
| Fair | 308 (20.5) | 84 (26.2) | 149 (18.5) | 75 (19.8) | |

Continued

**Table 2** Continued

| Characteristics | Total, n (%) | Low, n (%) | Moderate, n (%) | High, n (%) | P values |
|---|---|---|---|---|---|
| Good | 780 (51.8) | 173 (53.9) | 429 (53.2) | 178 (47.1) | |
| Excellent | 417 (27.7) | 64 (19.9) | 228 (28.3) | 125 (33.1) | 0.001* |
| Hypertension | | | | | |
| Yes | 205 (13.0) | 64 (18.7) | 107 (12.6) | 34 (8.8) | |
| No | 1376 (87.0) | 278 (81.3) | 744 (87.4) | 354 (91.2) | <0.001* |
| High blood cholesterol | | | | | |
| Yes | 183 (11.6) | 53 (15.3) | 95 (11.2) | 35 (9.1) | |
| No | 1397 (88.4) | 293 (84.7) | 754 (88.8) | 350 (90.9) | 0.028* |
| Heart attack | | | | | |
| Yes | 10 (0.6) | 2 (0.6) | 8 (0.9) | 0 (0.0) | |
| No | 1580 (99.4) | 346 (99.4) | 846 (99.1) | 388 (100.0) | 0.152 |
| Stroke | | | | | |
| Yes | 1 (0.1) | 0 (0.0) | 0 (0.0) | 1 (0.3) | |
| No | 1589 (99.9) | 348 (100.0) | 853 (100.0) | 388 (99.7) | 0.213 |
| Cancer | | | | | |
| Yes | 2 (0.1) | 2 (0.6) | 0 (0.0) | 0 (0.0) | |
| No | 1590 (99.9) | 346 (99.4) | 854 (100.0) | 390 (100.0) | 0.048* |
| Diabetes | | | | | |
| Yes | 179 (11.3) | 45 (13.0) | 99 (11.6) | 35 (9.0) | |
| No | 1409 (88.7) | 301 (87.0) | 754 (88.4) | 354 (91.0) | 0.213 |
| Stomach problems | | | | | |
| Yes | 127 (8.0) | 29 (8.3) | 74 (8.7) | 24 (6.2) | |
| No | 1464(92.0) | 319 (91.7) | 780 (91.3) | 365 (93.8) | 0.310 |
| Bowel problems | | | | | |
| Yes | 192 (12.1) | 42 (12.1) | 104 (12.2) | 46 (11.8) | |
| No | 1398 (87.9) | 304 (87.9) | 750 (87.8) | 344 (88.2) | 0.981 |
| Allergies | | | | | |
| Yes | 145 (9.1) | 25 (8.1) | 73 (8.5) | 44 (11.3) | |
| No | 1444 (90.9) | 319 (91.9) | 781 (91.5) | 344 (88.7) | 0.212 |
| Osteoporosis | | | | | |
| Yes | 44 (2.8) | 5 (1.4) | 24 (2.8) | 15 (3.9) | |
| No | 1540 (97.2) | 341 (98.6) | 826 (97.2) | 373 (96.1) | 0.136 |
| Fractures | | | | | |
| Yes | 46 (2.9) | 8 (2.3) | 19 (2.2) | 19 (4.9) | |
| No | 1545 (97.1) | 340 (97.7) | 834 (97.8) | 371 (95.1) | 0.027* |
| Psychiatric illness | | | | | |
| Yes | 12 (0.8) | 5 (1.4) | 5 (0.6) | 2 (0.5) | |
| No | 1579 (99.2) | 342 (98.6) | 849 (99.4) | 388 (99.5) | 0.245 |
| Polycystic ovarian syndrome | | | | | |
| Yes | 45 (2.8) | 9 (2.6) | 25 (2.9) | 11 (2.8) | |
| No | 1543 (97.2) | 339 (97.4) | 826 (97.1) | 378 (97.2) | 0.946 |

*Significant at α=0.05 level.
GCC, Gulf Cooperation Council.

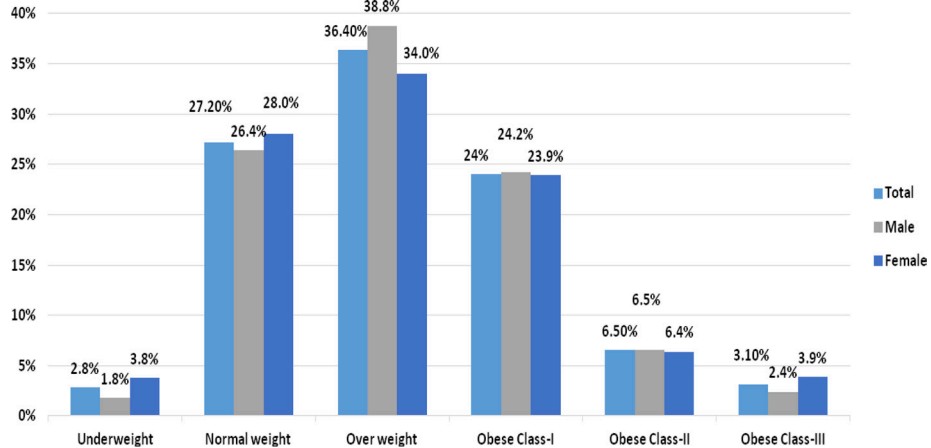

**Figure 2** Distribution of BMI among participants. BMI, body mass index.

Obesity class I, II and III were observed in 383 (24%), 103 (6.5%), and 50 (3.1%), respectively (figure 2).

Nearly one-third of the participants were obese. The majority of obese participants were classified as obese class I (24.2% men and 23.9% women). Thirty-four per cent of the female participants and 38.8% of the male participants were overweight. It is important to note that more women than men were in the underweight category and obese class III category based on their BMI.

### Level of PA

Sixty-four per cent of study participants did not perform any vigorous PA, and 39.9% did not perform any moderate PA during 7 days prior to the data collection (table 3). Within those 7 days, participants spent a mean (SD) of 31.12 (59.28) min doing vigorous PA, 46.87 (63.01) min doing moderate PA and 42.01 (47.04) min walking. Male participants spent significantly (p<0.001) more time doing vigorous PA (mean rank score 859) and walking (mean rank score 831.1) compared with female participants (mean rank score 690.6, mean rank score 730.5, respectively). However, men (mean rank score 769.5) and women (mean rank score 789.5) spent a similar amount

of time doing moderate PA (p=0.361). In this study, participants spent a mean (SD) of 7.39 (3.36) hours sitting per weekday. Approximately half of the participants (53.2%) spent 5–10 hours sitting, 30.3% spent 1–5 hours sitting and 16.2% spent 10–16 hours sitting per weekday. Only 0.3% of participants reported spending zero hours sitting per weekday. Sitting activity is classified as sedentary behaviour according to the Canadian Physical Activity and Sedentary Behaviour Guidelines.[24]

Most of the 1606 participants (98.6%) believed both PA and eating an HD were beneficial. However, less than half (45.4%) reported engaging in PA and only 34.5% reported eating healthy. Table 4 shows the participants' perceived level of PA and healthy eating.

Table 5 shows that for those categorised as physical activity engaged had a higher percentage of normal weight compared with those who did not engage (31.8% vs 20.6%). Similarly, those who ate healthy had a higher percentage of normal weight compared with those who did not eat healthy (30.0% vs 21.2%). For those 'sometimes' PA engaged and 'sometimes' eating healthy, the percentage of normal weight is shown in between 'Yes'

| Table 3 | Participants report of their physical activity level during 7 days (n=1606) | | | | | | | |
|---|---|---|---|---|---|---|---|---|
| Physical activities | None, n (%) | 1 day, n (%) | 2 days, n (%) | 3 days, n (%) | 4 days, n (%) | 5 days, n (%) | 6 days, n (%) | 7 days, n (%) | P values |
| Vigorous physical activities* | 987 (64) | 120 (7.8) | 123 (8) | 114 (7.4) | 49 (3.2) | 54 (3.5) | 23 (1.5) | 71 (4.6) | <0.001† |
| Moderate physical activities‡ | 622 (39.9) | 174 (11.2) | 186 (11.9) | 186 (11.9) | 45 (2.9) | 96 (6.2) | 38 (2.4) | 210 (13.5) | <0.001† |
| Walk for at least 10 min at a time§ | 272 (17.4) | 211 (13.5) | 226 (14.4) | 209 (13.3) | 68 (4.3) | 234 (14.9) | 79 (5) | 268 (17.1) | <0.001† |

All p values are computed by one sample $\chi^2$ test, with df=7.
*Sixty-five participants did not answer this question.
†Significant at α=0.05 level.
‡Forty-nine participants did not answer this question.
§Thirty-nine participants did not answer this question.

**Table 4** Level of physical activity and healthy eating (n=1604)

| Perceived level of physical activity and healthy eating | Yes, n (%) | No, n (%) | Sometimes, n (%) | P values |
|---|---|---|---|---|
| Do you engage in PA? | 729 (45.4) | 466 (29.1) | 409 (25.5) | <0.001* |
| Do you think you eat healthy food? | 553 (34.5) | 432 (26.9) | 618 (38.5) | <0.001* |

All p values are computed by one sample $\chi^2$ test, with df=2.
*Significant at $\alpha$=0.05 level.
PA, physical activity.

and 'No' groups. BMI category is statistically related to physical activity engagement and eating healthy, with both p<0.05.

### Dietary behaviour

Nearly one-third of the participants ate fresh fruits (35.8%), green vegetables (31.8%) and other vegetables such as carrots, tomatoes, cucumber (44.1%) at least once daily (table 6). Similarly, nearly one-third of the participants consumed protein products such as fish, beef or chicken 2–4 times weekly. One-quarter of the study participants drank full fat milk, 20.6% drank skimmed or low-fat milk and nearly 50% consumed milk products more than once daily. More than one-quarter of the study participants ate pasta and cakes or pastries 2–4 times in a week, 40.6% had a meal with white bread daily and 14.4% drank carbonated soda more than once daily.

Due to the ordinal nature of variables in tables 3, 4 and 6, Spearman rank correlation coefficients were computed to determine their linear relationships. The results reveal that there is little or no relationship (|r|<0.25) among the variables between tables 3 and 4 (except with physical activities engagement), tables 3 and 6 and tables 4 and 6. However, there is a fair degree of relationship between intensity of physical exercise (variables in table 3) and physical activities engagement (variable in table 4) with coefficients ranging from 0.298 to 0.485 (table 7).

Despite participants' perceived low level of being physically active (45.4%) and eating healthy (34.5%), the majority of participants found that engaging in PA (88.7%) and eating healthy foods (74.3%) were pleasant. Furthermore, 98.6% of participants believed that being active and eating healthy are beneficial for health. More than half strongly agreed that engaging in PA (73.1%) and eating health food (66.6%) would help maintain

good health. Out of 1606, 75.9% of participants communicated that they would engage in PA and 75.5% said they would eat healthy foods within the next month. Many were willing to continue to be physically active for the next 6 months (81.5%) and eat healthy foods (79.2%).

### Reported health status

Nearly half of the study participants (49.1%) reported good health, while the remaining cited excellent (26.2%), fair (19.3%) or poor (4.5%) health. Most participants did not have a medical history of heart attack, cancer, psychiatric illness, polycystic ovarian syndrome or fractures. Overall, 10%–13% of the participants reported a high blood cholesterol level, high blood pressure, diabetes, obesity or bowel problems (table 8).

For PA engagement prediction, a list of potential predictors: living area, marital status, nationality, age group, education, health status, household income and sex were chosen. Forward stepwise (Wald $\chi^2$, $p_{in}$=0.05, $p_{out}$=0.10) logistic regression method was used. Only marital status, age group, sex and education were statistically significant in predicting PA engagement.

Table 9 summarises multivariate forward stepwise logistic regression analyses results with significant independent factors that may be used as indicators to predict participant engagement in physical activities. Married participants had 0.676 times (p=0.021), older people had 0.679 times (p=0.009) and women had 0.348 times (p<0.001) the odds of physical activities engagements. Level of education was also a statistically significant predictor of physical activities engagements ($\chi2$ (5)=11.124, p=0.049).

Similarly, for eating healthy food, we used the same list of potential predictors as in predicting PA engagement. Variables such as living area, marital status, nationality, age group, education, health status, household income

**Table 5** Association between BMI and physical activity engagement and eating habits

| BMI category | >Physical activity engagement | | | | Eating healthy | | | |
|---|---|---|---|---|---|---|---|---|
| | Yes, n (%) | No, n (%) | Sometimes, n (%) | P values | Yes, n (%) | No, n (%) | Sometimes, n (%) | P values |
| Underweight | 18 (2.5) | 10 (2.2) | 16 (3.9) | | 17 (3.1) | 8 (1.9) | 19 (3.1) | |
| Normal | 230 (31.8) | 95 (20.6) | 107 (26.4) | | 164 (30.0) | 91 (21.2) | 178 (28.9) | |
| Overweight | 284 (39.3) | 167 (36.1) | 128 (31.5) | | 206 (37.7) | 155 (36.1) | 217 (35.3) | |
| Obese | 191 (26.4) | 190 (41.1) | 155 (38.2) | <0.001* | 160 (29.3) | 175 (40.8) | 201 (32.7) | 0.003* |

*Significant at $\alpha$=0.05 level.
BMI, body mass index.

**Table 6** Food intake for daily meal (n=1606)

| Food | Never n (%) | Seldom n (%) | 1/week n (%) | 2–4/week n (%) | 5–6/week n (%) | 1+ daily n (%) | P values |
|---|---|---|---|---|---|---|---|
| **Fruits and vegetables** | | | | | | | |
| Fresh fruits (apples, oranges, pears)* | 20 (1.3) | 123 (7.7) | 165 (10.3) | 469 (29.3) | 251 (15.7) | 572 (35.8) | <0.001† |
| Green leafy vegetables‡ (lettuce, cabbage, spinach) | 35 (2.2) | 99 (6.2) | 180 (11.2) | 485 (30.3) | 293 (18.3) | 510 (31.8) | <0.001 |
| Other vegetables, carrots, tomatoes, cucumber)§ | 22 (1.4) | 68 (4.2) | 85 (5.3) | 392 (24.4) | 326 (20.3) | 708 (44.1) | <0.001† |
| **Proteins** | | | | | | | |
| Fish¶ | 128 (8) | 292 (18.3) | 609 (38.1) | 462 (28.9) | 63 (3.9) | 45 (2.8) | <0.001† |
| Beef** | 354 (22.3) | 352 (22.1) | 334 (21) | 417 (26.2) | 78 (4.9) | 56 (3.5) | <0.001† |
| Lamb†† | 197 (12.4) | 279 (17.6) | 397 (25.1) | 532 (33.6) | 91 (5.7) | 88 (5.6) | <0.001† |
| Chicken* | 18 (1.1) | 38 (2.4) | 177 (11.1) | 737 (46.1) | 345 (21.6) | 285 (17.8) | <0.001† |
| Meat products (sausages, burgers, shawarma)‡‡ | 276 (17.3) | 496 (31.1) | 341 (21.4) | 305 (19.1) | 110 (6.9) | 67 (4.2) | <0.001† |
| Eggs‡‡ | 102 (6.4) | 209 (13.1) | 277 (17.4) | 580 (36.4) | 176 (11) | 251 (15.7) | <0.001† |
| Legumes (lentils, beans, peas)§§ | 102 (6.4) | 284 (17.7) | 391 (24.3) | 545 (33.9) | 136 (8.5) | 128 (8) | <0.001† |
| **Milk Products** | | | | | | | |
| Milk, full fat** | 430 (27) | 234 (14.7) | 115 (7.2) | 245 (15.4) | 174 (10.9) | 393 (24.7) | <0.001† |
| Milk, low fat or skimmed¶¶ | 528 (33.1) | 317 (19.9) | 104 (6.5) | 199 (12.4) | 120 (12.5) | 328 (20.6) | <0.001† |
| Milk product (cheese, yoghourt, milk drinks)*** | 38 (2.4) | 90 (5.6) | 100 (6.2) | 393 (24.5) | 266 (16.6) | 716 (44.7) | <0.001† |
| **Carbohydrates** | | | | | | | |
| Bread, white§ | 169 (10.6) | 175 (10.9) | 81 (5.1) | 257 (16.1) | 237 (14.8) | 682 (42.6) | <0.001† |
| Bread, whole meal/brown††† | 432 (27) | 408 (25.5) | 131 (8.2) | 201 (12.6) | 95 (5.9) | 331 (20.7) | <0.001† |
| Cereals (cornflakes, oatmeal)‡‡‡ | 494 (30.9) | 423 (26.5) | 225 (14.1) | 208 (13) | 75 (4.7) | 172 (10.7) | <0.001† |
| Pasta (spaghetti, macaroni, noodles, grits)*** | 72 (4.5) | 201 (12.5) | 387 (24.1) | 469 (29.3) | 202 (12.6) | 272 (17) | <0.001† |
| Snack foods (potato chips, popcorn, chocolates…)* | 211 (13.2) | 385 (24.1) | 307 (19.2) | 368 (23) | 135 (8.4) | 194 (12.1) | <0.001† |
| Cakes and pastries (cakes, biscuits, sweet pies)* | 143 (8.9) | 414 (25.8) | 351 (21.9) | 416 (26) | 126 (7.9) | 150 (9.4) | <0.001† |
| **Others** | | | | | | | |
| Soft drinks (cola drinks)§ | 531 (33.2) | 335 (20.9) | 171 (10.7) | 223 (13.9) | 111 (6.9) | 230 (14.4) | <0.001† |
| Nuts (pistachio, cashew nuts)§§§ | 154 (9.7) | 501 (31.5) | 335 (21.1) | 346 (21.8) | 79 (5) | 174 (11) | <0.001† |
| Fruit Juices¶ | 102 (6.4) | 217 (13.6) | 263 (16.4) | 508 (31.8) | 204 (12.8) | 305 (19.1) | <0.001† |
| Fruit drinks* | 310 (19.4) | 321 (20.1) | 131 (8.2) | 373 (23.3) | 158 (9.8) | 307 (19.1) | <0.001† |
| Coffee/tea§ | 77 (4.8) | 82 (5.1) | 51 (3.2) | 142 (8.9) | 216 (13.5) | 1033 (64.5) | <0.001† |

All p values are computed by one sample $\chi^2$ test, with df=5.
*Six participants did not answer this question.
†Significant at $\alpha$=0.05 levels.
‡Four participants did not answer this question.
§Five participants did not answer this question.
¶Seven participants did not answer this question.
**Fifteen participants did not answer this question.
††Twenty-two participants did not answer this question.
‡‡Eleven participants did not answer this question.
§§Twenty participants did not answer this question.
¶¶Ten participants did not answer this question.
***Three participants did not answer this question.
†††Eight participants did not answer this question.
‡‡‡Nine participants did not answer this question.
§§§Seventeen participants did not answer this question.

and sex were selected. Forward stepwise (Wald $\chi^2$, $p_{in}$=0.05, $p_{out}$=0.10) logistic regression method was used. Only age group, health status and living area were statistically significant in predicting eating healthy food.

Table 10 summarises multivariate forward stepwise (Wald $\chi^2$, $p_{in}$=0.05, $p_{out}$=0.10) logistic regression analyses results with significant independent factors that may be used as indicators to predict participants eating healthy

**Table 7** Spearman rank correlation coefficients between physical activities engagement and intensity of physical exercise

| | Vigorous physical exercise | Moderate physical exercise | Walk for at least 10 min at a time |
|---|---|---|---|
| Physical activities engagement | 0.485 (p<0.001*, n=1539) | 0.298 (p<0.001*, n=1555) | 0.348 (p<0.001*, n=1565) |

*Significant at α=0.05 levels.

food. Older people had 0.68 times (p=0.005) and people who lived in Al Wakrah (South of Qatar) had 0.57 times (p=0.007) the odds of eating healthy food. Living area is a statistically significant predictor of eating healthy food ($\chi^2$ (2)=8.35, p=0.007). Health status is also a statistically significant predictor with one unit improvement in health status having 1.35 times (p=0.002) the odds of eating healthy food.

For IPAQ groups prediction, due to the ordinal nature of IPAQ groups, ordinal logistic regression was supposed to be used to determine the relationship between IPAQ groups and the following variables: living area, marital status, nationality, age group, education, health status, household income and sex. However, it did not satisfy the assumption of parallel lines for ordinal logistic regression (−2 log likelihood=1728.24, $\chi^2$(22)=52.90, p<0.001). Therefore, we combined moderate and high IPAQ groups into one group. Forward stepwise (Wald $\chi^2$, $p_{in}$=0.05, $p_{out}$=0.10) logistic regression was performed on

two groups (group 1, low; group 2, moderate and high combined).

Table 11 summarises forward stepwise logistic regression results in predicting 'moderate and high combined'. Out of this set of potential predictors, only marital status and health status were statistically significant in predicting moderate and high IPAQ groups. Married participants had 0.75 times (p=0.042) the odds of 'moderate and high' and one unit improvement in health status which had 1.41 times (p=0.001) the odds of having 'moderate and high' PA.

## DISCUSSION

In this cross-sectional community-based survey study, we examined the level of PA, dietary habits and health status among adult Arabic speakers living in the state of Qatar. Our findings highlight the low PA levels in study participants. It is known that low PA level in the Middle East

**Table 8** Medical history of study participants (n=1606)

| Medical history | Yes, n (%) | No, n (%) | P values |
|---|---|---|---|
| High blood pressure* | 206 (12.9) | 1389 (87.1) | <0.001† |
| Bowel problems‡ | 195 (12.2) | 1409 (87.8) | <0.001† |
| High blood cholesterol§ | 183 (11.5) | 1411 (88.5) | <0.001† |
| Diabetes¶ | 180 (11.2) | 1422 (88.8) | <0.001† |
| Obesity** | 162 (10.1) | 1443 (89.9) | <0.001† |
| Allergies†† | 146 (9.1) | 1457 (90.9) | <0.001† |
| Stomach problems** | 130 (8.1) | 1475 (91.9) | <0.001† |
| Asthma** | 73 (4.5) | 1532 (95.5) | <0.001† |
| Fractures** | 46 (2.9) | 1559 (97.1) | <0.001† |
| Polycystic ovarian syndrome¶ | 45 (2.8) | 1557 (97.2) | <0.001† |
| Osteoporosis‡‡ | 44 (2.8) | 1553 (97.2) | <0.001† |
| Psychiatric illness** | 12 (0.7) | 1593 (99.3) | <0.001† |
| Heart attack‡ | 10 (0.6) | 1594 (99.4) | <0.001† |
| Cancer | 2 (0.1) | 1604 (99.9) | <0.001† |
| Stroke‡ | 1 (0.1) | 1603 (99.9) | <0.001† |

All p values are computed by one sample $\chi^2$ test, with df=1.
*Eleven participants did not answer this question.
†Significant at α=0.05 levels.
‡Two participants did not answer this question.
§Twelve participants did not answer this question.
¶Four participants did not answer this question.
**One participant did not answer this question.
††Three participants did not answer this question.
‡‡Nine participants did not answer this question.

**Table 9** Association between significant factors (results from forward stepwise logistic regression) and physical activities engagement

| | Crude OR (95% CI) | P values | Adjusted OR (95% CI) | P values |
|---|---|---|---|---|
| Predictors of physical activities engagement | | | | |
| Marital status | | | | |
| Not married (reference) | | | 1.0 | |
| Married | 0.54 (0.40 to 0.71) | <0.001* | 0.68 (0.49 to 0.94) | 0.021* |
| Age group | | | | |
| 18–39 (reference) | 1.0 | | 1.0 | |
| 40 and older | 0.57 (0.44 to 0.74) | <0.001* | 0.68 (0.51 to 0.91) | 0.009* |
| Sex | | | | |
| Male (reference) | 1.0 | | 1.0 | |
| Female | 0.38 (0.29 to 0.49) | <0.001* | 0.35 (0.27 to 0.46) | <0.001* |
| Education of participant (Wald $\chi^2(5)=11.12$) | | | | 0.049* |
| ≤Primary/intermediate school (reference) | 1.0 | | 1.0 | |
| Primary/junior high | 0.72 (0.20 to 0.26) | 0.607 | 0.71 (0.19 to 2.65) | 0.615 |
| High school | 1.51 (0.45 to 5.12) | 0.508 | 1.04 (0.29 to 3.72) | 0.949 |
| Trade | 0.39 (0.90 to 1.65) | 0.090 | 0.25 (0.06 to 1.13) | 0.072 |
| University | 1.03 (0.31 to 3.47) | 0.961 | 0.82 (0.23 to 2.89) | 0.761 |
| Other | 1.19 (0.34 to 4.20) | 0.792 | 0.98 (0.27 to 3.62) | 0.978 |

Model summary: −2 log likelihood=1333.527; Cox & Snell $R^2$=0.080; Nagelkerke $R^2$=0.113.

is related to its weather condition,[25 26] that is, the environmental factor of a hot desert climate is a barrier for people to engage in exercise and participate in outdoor activities. In summer, the temperature is very high in Qatar (30° C–50°C), which restricts outdoor activities such as walking, cycling and jogging. Other reported barriers include lack of interest, motivation and information about the benefits of exercise, stress, excessive internet and computer usage, and lack of accessible exercise facilities.[27 28]

Congruent with previous research and surveys done in Qatar and GCC countries, which were discussed in the beginning of this report, the findings of this study suggests that the Qatar adult population's PA is insufficient to meet the amount recommended by the WHO and Qatar Physical Activity Guidelines. In addition, it was noted that there is a difference in the level of vigorous (ie, running) and mild (ie, casual walking) activities between male and female participants. Previous studies have reported similar findings with women being less involved in regular PA.[29 30] Similarly, recent work in Qatar revealed that men were more physically active than women, both for walking (74.6% of men vs 55.3% of women) and intense sports (16.5% of men vs 8.9% of women).[31] An Omani study found that female participants were significantly more inactive compared with male participants (76.9% vs 33.3%).[32] Direct comparisons across studies are complicated due to multiple factors such as methodological

**Table 10** Association between significant factors (results from forward stepwise logistic regression) and 'Do you think you eat healthy food?'

| Predictors of 'Do you think you eat healthy food?' | Crude OR (95% CI) | P values | Adjusted OR (95% CI) | P values |
|---|---|---|---|---|
| Age group | | | | |
| 18–39 (reference) | 1.0 | | 1.0 | |
| 40 and older | 0.68 (0.52 to 0.89) | 0.004* | 0.68 (0.52 to 0.89) | 0.005* |
| Living area (Wald $\chi^2(2)=8.09$) | | | | 0.018* |
| Al Doha (reference) | 1.0 | | 1.0 | |
| Al Wakrah (South of Qatar) | 0.57 (0.38 to 0.86) | 0.008* | 0.57 (0.38 to 0.85) | 0.007* |
| Al Khor/Al Thakhira (North of Qatar) | 1.10 (0.74 to 1.63) | 0.632 | 1.13 (0.76 to 1.68) | 0.558 |
| Health status | 1.35 (1.12 to 1.63) | 0.002* | 1.35 (1.11 to 1.63) | 0.002* |

Model summary: −2 log likelihood=1315.497; Cox & Snell $R^2$=0.021; Nagelkerke $R^2$=0.031.

variations, ethnicity, geographical location and cultural values.[33] However, the low level of involvement of women in regular physical activities in Qatar is possibly related to cultural and social norms (ie, women in many Islamic countries need to be accompanied by a male family member when going outdoors), caregiving responsibilities, the need to wear an abaya in public and a general lack of social support for exercise.[8 25 28 34–37] Even though the present study's data indicated that female participants were less likely to be physically active, older and married women had slightly higher odds of being involved in PA. We anticipated that this might be related to the fact that older married women were more physically involved in family domestic activities such as house chores and taking care of children and other family members.

Sitting has been highlighted as a specific marker of sedentary behaviour.[38] In this study, it was alarming that slightly more than half (53.2%) of the participants spent 5–10 hours sitting per day. In contrast, a recently published research article reported that the overall median time spent in sedentary activities was 179 min (2.98 hours) per day among Qatari nationals.[26] This discrepancy in findings is perhaps due to diverse participant characteristics. Our study has more participants with different ethnocultural backgrounds than just Qatari nationals. More sitting time and insufficient PA are significantly associated with higher health risk factors such as abdominal obesity, dyslipidaemia, hyperglycaemia and hypertension among the adult population.[39]

In the present study, about two-thirds of the participants were either overweight or obese with a mean BMI of $28.03 \, \text{kg/m}^2$. Our findings are consistent with a previous study conducted in Qatar, where 70.1% of the participants had a BMI equal to or above $25 \, \text{kg/m}^2$ (classified as overweight and obese).[31] The prevalence of obesity in GCC countries is among the highest in the world. According to the Global Status Report in 2014,[10] rates of obesity reached more than 37% in the UAE, almost 40% in Kuwait and more than 42% in Qatar. Similarly, high BMI in Saudi Arabia (34.7%) and Lebanon (31.9%) have been reported.[40]

The rapid increase in wealth and subsequent development of Arab countries has led to changes in lifestyle. One of the many consequences of region development secondary to abundant oil resources has been a significant change in eating habits. The GCC's former Mediterranean-style diet of traditional products (eg, dates, vegetables, wheat) has been replaced by a reliance on fast foods that are dominated by refined and processed meals heavy in sugar and other carbohydrates.[41] This might help to explain our findings that older participants who lived outside of the capital had slightly higher odds of eating healthy because they might be less likely to adopt or access Western-style diet. There is strong evidence of the benefits of HD in primary prevention of major cardiovascular events and reducing the risk of diabetes among people with high cardiovascular risk.[42 43] In a recent qualitative study, Donnelly and colleagues[28] examined sociocultural factors that influenced the healthy lifestyles (ie, diet, PA, non-smoking) of 50 Arabic women with heart diseases living in Qatar. The participants reported that their diets tended to be high in salt, sugar and fats. Their diets were also influenced by traditional cultural beliefs and values. For example, women were often invited to each other's homes where sweets and coffee were served. The women ate foods that were offered because refusing to eat the food would be considered socially unacceptable behaviour. Furthermore, Donnelly and colleagues[34] reported that social support, cultural values, religion, hot desert climate, heart disease, changing sociodemographic and economic conditions impacted both positively and negatively on the ability of these women to pursue a healthy lifestyle.

Overall, participants' dietary habits in this study were found to be unhealthy: 40.6% ate white bread daily and had pasta and cakes or pastries 2–4 times per week. Non-carbohydrate-containing foods such as fruits, legumes, vegetables and minimally processed whole grains are healthy and cardiometabolic protective, while foods rich in refined grains (eg, white bread, white rice), crackers, cereals, bakery desserts, starches and added sugars are associated with weight gain.[20] A study conducted in Oman reported few gender differences with respect to eating habits, except in dairy and meat consumption where 62.5% and 55.5% of men consumed more than three servings, compared with 18.78% and 35.2% of women, respectively.[32] Dietary behaviours are mainly influenced by individual factors (taste preferences, dietary knowledge, stress, body image, former eating habits and PA level) and environmental factors (availability and accessibility, costs of food products).[44] Given that more women than men were in the categories

**Table 11** Association between significant factors (results from forward stepwise logistic regression on IPAQ group (low vs 'moderate and high')

| Predictors of IPAQ (low vs 'moderate and high') | Crude OR (95% CI) | P values | Adjusted OR (95% CI) | P values |
|---|---|---|---|---|
| Marital status | | | | |
| Not married (reference) | 1.0 | | 1.0 | |
| Married | 0.75 (0.54 to 1.02) | 0.069 | 0.72 (0.52 to 0.99) | 0.042* |
| Health status | 1.41 (1.15 to 1.74) | 0.001* | 1.43 (1.16 to 1.77) | 0.001*0.001* |

Model summary: −2 log likelihood=1145.37; Cox & Snell $R^2$=0.013; Nagelkerke $R^2$=0.020.

of underweight and obese class III in our study, facilitating Arab female engagement in healthier lifestyle needs to be emphasised. There is a need to develop collaboration among healthcare professionals, academics, public health professionals and policy-makers to improve the dietary behaviours and lifestyle of the population. In the UK, the National Institute for Health and Care Excellence recommends that primary care practitioners deliver brief PA advice to lethargic adults and follow up on outcomes at subsequent appointments.[45]

## CONCLUSION

We conclude that there are insufficient levels of PA and poor dietary behaviour among the population in Qatar. Insufficient levels of PA and unhealthy dietary present a great health risk. There is a need to develop a nationwide health promotion programme that aims at increased knowledge about the benefits of being physically active and eating healthy. Awareness of and adherence to Qatar Physical Activity Guidelines and the new Arabic dietary guidelines should be encouraged by healthcare policy-makers and healthcare providers. The findings of this study might provide insights and information necessary for the development of public health policies and promotion programmes in Qatar and in the Middle Eastern region.

## Study limitations

The present study nonetheless has multiple limitations that should be considered when interpreting the results. Non-probability convenience sampling limits the ability to generalise the findings from this study. In addition, the findings of this study might not be valid for generalisation to other population segments of different race/ethnicity and socioeconomic status due to geographic/cultural differences. However, the findings are relevant to a population with similar ethnic and cultural backgrounds. In this study, participants were 18 years of age and over. Younger participants (age <18 years) might possibly give different perspectives with regard to their PA and dietary behaviours. In addition, information on dietary habits in the present study was based on the frequency of consumption of food items without much consideration to quantity or portion size. Future research is needed to investigate the association between Arabs' attitude, perceived behavioural control and their PA level and dietary habit.

**Acknowledgements** We thank our study participants who accepted to join us and share their experiences. We hope through this report we repay them for their insight and trust. We give special thanks to our research team, to the staff at universities, colleges and primary health care centres who helped us to recruit the research participants, to our project coordinator, Shima Sharara and to our research assistants: Nagla HagOmer, Sara Saeed and Samira Mouhamed. We also thank Muhammad Siddiqui for his assistance with this report.

**Contributors** TTD: contributed to the conception and design of the study and the acquisition, analysis and interpretation of data, drafted the manuscript and gave final approval of the manuscript version submitted for publication. TSF: contributed to the conception and design of the study and the acquisition, analysis

and interpretation of data, revised the manuscript and gave final approval of the manuscript version submitted for publication. A-AbMA-T: contributed to the conception and design of the study and the acquisition of data, reviewed the manuscript critically for content, and gave final approval of the manuscript version submitted for publication.

**Funding** This study was made possible by a grant from the Qatar National Research Fund under its National Priorities Research Program (NPRP 6–049–3–009).

**Disclaimer** The contents of this study are solely the responsibility of the authors and do not necessarily represent the official views of the Qatar National Research Fund.

**Competing interests** None declared.

**Patient consent** Obtained.

**Ethics approval** The ethical approvals were obtained from several ethics review boards (IRBs) which include: Hamad Medical Corporation/Weill Cornell Institutional Review Board (IRB NO: 13-00051), the University of Calgary's Conjoint Health Research Ethics Board (REB13-0347), Qatar Primary Health Care Research Committee (Reference NO: PHCC/RC/14/02/004), Qatar University (QU-IRB 244-E/13), College of North Atlantic in Qatar (CNAQ Approval NO: 2015-3) and Qatar Supreme Council of Health (SCH-A-UCQ-050).

**Provenance and peer review** Not commissioned; externally peer reviewed.

**Data sharing statement** No additional data are available.

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
