## [Reviewer comments · BMJ Open]

ARTICLE DETAILS

TITLE (PROVISIONAL)	Fostering active living and healthy eating through understanding physical activity and dietary behaviours of Arabic-speaking adults – a cross-sectional study from the Middle East
AUTHORS	Donnelly, Tam Truong; Fung, Tak; Al-Thani, Al-Anoud

VERSION 1 – REVIEW

REVIEWER	Professor Sharon Brownie Dean, School of Nursing and Midwifery in East Africa Aga Khan University Nairobi Kenya
REVIEW RETURNED	25-Oct-2017

GENERAL COMMENTS	Rising rates of obesity and non-communicable disease are a very significant challenge to the modern Arab world. This article provides insight to dietary and exercise patterns of the adult population in Qatar. The study involves a simple methodology resulting in descriptive analysis of current patterns and behaviors. I am not sure it adds much more to what is generally already known in Qatar and across the Middle Eastern region in general. Never-the-less the scale of the study provides insight to the quantum of the problem and is a useful piece of work from that perspective. English and grammar is generally good, however, there are items scattered through script which would require a final edit and correction prior to publication, for example,  • Line 28 page 2 – ‘third’ should be ‘thirds’ • Line 21 page 3 – ‘report’ should be ‘reports’ • Line 4 page 4 – ‘chronic diseases preventative efforts’ would better read ‘chronic disease prevention efforts’
--

REVIEWER	Marcia Erazo Universidad de Chile Chile
REVIEW RETURNED	30-Oct-2017

GENERAL COMMENTS	In the introduction section, lines 17-24, authors mix AMI's risk factors and protector factors, please provide the risk factors only (i.e., physical activity / sedentary lifestyle; daily consumption of fruit and vegetables/ occasional-non-daily consumption of v&f). The authors declare that they did not conduct a randomized sampling. Please explain why they chose a non-probability sampling, and collected data from universities and health care
---

	centers, where population is probably not very representative of the general population. Please explain the parameters used to estimate the sample size. During the data collection, authors declare that they determined "dietary behavior" using a questionnaire. Please explain if your questionnaire was a 24-h recall, a food consumption tendency questionnaire, or other. Considering the type of the food consumption survey, please provide the methodological details that you kept to capture the actual food consumption of your population. In the results section, you express that 1606 participants met the "inclusion criteria", nevertheless they are not expressed in the methods, please provide them. Please provide a figure, explaining the number of initially selected persons, the number of those who did not match the inclusion criteria, the number of those who rejected to participate in the study, and the final sample size. In the methods section, please explain the way you chose and included the participants, especially when they rejected or did not match the inclusion criteria. Table 1, table 2, table 3, table 4, table 5, please provide p-value for the variables reported. Considering the information provided in tables 2 and 4, authors could create an index to reflex the level of "healthy eating" and "healthy physical activity" levels of the population, and contrast them with data reported by population itself in table 3. It would be interesting that authors could build a regression model that explains a "poor healthy eating and physical activity" level (including sociodemographic variables). In the discussion section the authors compare several studies with the results provided by this one, nevertheless, they never discuss the limitations of this study, especially considering that: 1 is a cross-sectional design, and 2: a non-randomized sample study. Please include in this section a paragraph discussing the limitations, focusing on the bias of this study. In the conclusion section, the authors state that this study provides information to develop public health policies, it would have been, if the sampling were randomized and nationwide. Please rephrase the conclusion considering the methodological limitations and bias of this study.
--	--

REVIEWER	Lena Zimmo, PhD Aspetar- Orthopaedic and Sports Medicine Hospital Sport City Street Near Khalifa Stadium P.O. Box 29222 Doha, Qatar
REVIEW RETURNED	06-Nov-2017

GENERAL COMMENTS	I enjoyed reading your paper, great effort. However, the final regression analysis is missing. The regression analysis can be logistic regression for determining how age, gender, nationality, and level of income (as shown in table 1) are associated with  1. Physical activity 2. Dietary habit. Doing the regression analysis will enrich the result and the discussion. Looking forward to read you paper and I am sure it will look great and very informative. -The reviewer also provided a marked copy with additional comments. Please contact the publisher for full details.
--

VERSION 1 – AUTHOR RESPONSE

Reviewer: 1

Reviewer Name: Professor Sharon Brownie

Institution and Country: Dean, School of Nursing and Midwifery in East Africa, Aga Khan University, Nairobi, Kenya

Comments:

Rising rates of obesity and non-communicable disease are a very significant challenge to the modern Arab world. This article provides insight to dietary and exercise patterns of the adult population in Qatar.

The study involves a simple methodology resulting in descriptive analysis of current patterns and behaviors. I am not sure it adds much more to what is generally already known in Qatar and across the Middle Eastern region in general. Never-the-less the scale of the study provides insight to the quantum of the problem and is a useful piece of work from that perspective.

English and grammar is generally good, however, there are items scattered through script which would require a final edit and correction prior to publication, for example,

- Line 28 page 2 – ‘third’ should be ‘thirds’
- Line 21 page 3 – ‘report’ should be ‘reports’
- Line 4 page 4 – ‘chronic diseases preventative efforts’ would better read ‘chronic disease prevention efforts’

Response: Thank you for your comments. We have revised the paper based on your suggestions.

Reviewer: 2

Reviewer Name: Marcia Erazo

Institution and Country: Universidad de Chile, Chile

Comments:

In the introduction section, lines 17-24, authors mix AMI's risk factors and protector factors, please provide the risk factors only (i.e., physical activity / sedentary lifestyle; daily consumption of fruit and vegetables/ occasional-non-daily consumption of v&f).

Response: Thank you for your comments. We have changed the sentence as per your advice.

Comments: The authors declare that they did not conduct a randomized sampling. Please explain why they chose a non-probability sampling, and collected data from universities and health care centers, where population is probably not very representative of the general population.

Response: We inserted our explanation on page 6 as follow “Although random selection helps to reduce selection biases, this sampling technique would not be feasible for this research. It is difficult to gain access to Arabic populations because social and cultural beliefs and practices values privacy. Hence, we used a purposeful, non-probability convenient sampling technique. We realized that this will increase the risk of selection biases and will limit the generalizability of the research findings. To help offset these limitations, we recruited and randomly selected participants at different times of the days, weeks, or months of the year at the designated data collection sites (universities and community health clinics).”

Comments: Please explain the parameters used to estimate the sample size.

Response: On page 7, we have indicated “Based on Cochran’s formula for sample size determination, a sample size of 781 women and 784 men to obtain a representative sample for

843,441 men and 165,496 females aged 15 to 75+ living in the three regions mentioned above, and using a margin of errors of 3.5% (95% confidence interval) [20].” We have also added more information explaining the participant’s recruitment process.

Comments: During the data collection, authors declare that they determined "dietary behavior" using a questionnaire. Please explain if your questionnaire was a 24-h recall, a food consumption tendency questionnaire, or other. Considering the type of the food consumption survey, please provide the methodological details that you kept to capture the actual food consumption of your population.

Response: on page 9 we added: “For example, question was asked: How often do you eat fresh fruits? Never; Seldom; Once a week; 2-4 times a week; 5-6 times a week; Once or more daily; Don’t know.”

Comments: In the results section, you express that 1606 participants met the "inclusion criteria", nevertheless they are not expressed in the methods, please provide them. Please provide a figure, explaining the number of initially selected persons, the number of those who did not match the inclusion criteria, the number of those who rejected to participate in the study, and the final sample size.

Response: on page 10 we added: “Three thousand and eighty one (n=3081) participants were approached of which 1606 participants who met the study’s inclusion criteria participated in the study (response rate 52.1%).”

Comments: In the methods section, please explain the way you chose and included the participants, especially when they rejected or did not match the inclusion criteria.

Response: On page 6 and 7, we added: “Trained interviewers who are fluent in both Arabic and English identified eligible participants based on the inclusion criteria. Providing the eligible individual answers “yes” to the screening questions and wishes to continue, the interviewer: (1) provided the participant with a short explanation of the study, (2) advised the participant that his/her participation is strictly voluntary, (3) advised him/her that measures that will be taken to help ensure confidentiality, and (4) answered any of the participant’s questions. If the participant agreed to participate, this was considered “consent by assent”. The interviewer then enrolled the participant in the study and administered the survey. Recruitment continued until determined sample size was reached.”

Comments: Table 1, table 2, table 3, table 4, table 5, please provide p-value for the variables reported. Considering the information provided in tables 2 and 4, authors could create an index to reflex the level of "healthy eating" and "healthy physical activity" levels of the population, and contrast them with data reported by population itself in table 3. It would be interesting that authors could build a regression model that explains a "poor healthy eating and physical activity" level (including sociodemographic variables).

Response: We have added p-value to tables: 1, 2, 3, 4. 6. On page 18 we added the below and table 5.

Due to ordinal nature of variables in Tables 2, 3 & 4, Spearman rank correlation coefficients are computed to determine their linear relationships. Results reveal that there is little or no relationship ($|r| < .25$) among the variables between Tables 2 & 3 (except with Physical Activities Engagement), 2 & 4, 3 & 4. However, there is a fair degree of relationship between variables in Table 2 with Physical Activities Engagement in Table 3 with coefficients ranges from 0.298 to 0.485.

Table 5. Spearman Rank Correlation Coefficients between Physical Activities Engagement and Intensity of Physical Exercises

Physical Activities Engagement Vigorous physical exercise Moderate physical exercise Walk for at least 10 minutes at a time

0.485(p<.001*,n=1539) 0.298(p<.001*, n=1555) 0.348(p<.001*,n=1565)
 *significant at $\alpha=0.05$ levels.

On page 21, we added the below:

Table 7 summarizes multivariate logistic regression analyses results performed with selected independent factors that may be used as indicators to predict participant engaged in physical activities. Married had 0.68 times ($p=0.020$), older people had 0.65 times ($p=0.002$) and female had 0.684 times ($p<0.001$) of the odds of physical activities engagements. Highest level of education is also a statistical significant predictor of physical activities engagements with $\chi^2(5) = 11.73$, $p=0.039$.

Table 7. Association between Selected Factors and Physical Activities engagement

Predictors of Physical Activities Engagement	Adjusted OR (95% CI)	P value
Marital Status		
Not married (reference)	1.0	
Married	0.68 (0.50 – 0.94)	0.020*
Agegroup		
18-39 (reference)	1.0	
40 & older	0.65 (0.49 – 0.86)	0.002*
Gender		
Male (reference)	1.0	
Female	0.38 (0.29 – 0.49)	<0.001*
Education of Participant (Wald $\chi^2(5)=11.73$)		
\leq Primary/intermediate school(reference)	1.0	0.039*
Primary/Junior High	0.69 (.19 – 2.53)	0.571
High School	0.94 (.26 - 3.32)	0.920
Trade	0.22 (.05 - .97)	0.045*
University	0.77 (.22 - 2.70)	0.687
Other	0.88 (.24 – 3.22)	0.849

Model summary

-2 Log likelihood	Cox & Snell R Square	Nagelkerke R Square
1426.38	0.08	0.11

On page 22, we added the below:

Table 8 summarizes multivariate logistic regression analyses results performed with selected independent factors that may be used as indicators to predict participant eating healthy food. Older people had 0.66 times ($p=0.001$) and people live in Al-Wakra (South of Qatar) had 0.66 times

(p=0.019) of the odds of eating healthy food. Live area is also a statistical significant predictor of eating healthy food with $\chi^2(2) = 8.09$, $p=0.018$.

Table 8. Association between Selected Factors and 'do you think you eat healthy food?'

Predictors of 'do you think you eat healthy food?'	Adjusted OR (95% CI)	P value
Age group		
18-39 (reference)	1.0	
40 & older	0.66 (0.51 – 0.85)	0.001*
Live Area(Wald $\chi^2(2)=8.09$)		
Al-Doha (reference)	1.0	0.018*
Al-Wakra (South of Qatar)	0.62 (0.41 – 0.92)	0.019*
Al-Khor/ Al Thakhira (North of Qatar)	1.29 (0.87 – 1.90)	0.204

Model summary

-2 Log likelihood	Cox & Snell R Square	Nagelkerke R Square
1431.66	0.02	0.02

Comments: In the discussion section the authors compare several studies with the results provided by this one, nevertheless, they never discuss the limitations of this study, especially considering that: 1 is a cross-sectional design, and 2: a non-randomized sample study. Please include in this section a paragraph discussing the limitations, focusing on the bias of this study.

Response: In addition to our explanation regarding the method of participants' selection on page 6, we added the following to the section "Study limitations": Non-probability convenience sampling limits the ability to generalise the findings from this study.

Comments: In the conclusion section, the authors state that this study provides information to develop public health policies, it would have been, if the sampling were randomized and nationwide. Please rephrase the conclusion considering the methodological limitations and bias of this study.

Response: We rephrased the sentence as requested.

Reviewer: 3

Reviewer Name: Lena Zimmo, PhD

Institution and Country: Aspetar- Orthopaedic and Sports Medicine Hospital, Sport City Street, Near Khalifa Stadium, Doha, Qatar

Comments:

I enjoyed reading your paper, great effort. However, the final regression analysis is missing. The regression analysis can be logistic regression for determining how age, gender, nationality, and level of income (as shown in table 1) are associated with

1. Physical activity
2. Dietary habit.

Doing the regression analysis will enrich the result and the discussion.
Looking forward to read your paper and I am sure it will look great and very informative.

Response: Thank you very much for your positive comments. We have added more information on page 18, 21, and 22.

We have also added to the discussion section (page 24) the below sentences to reflect added information on table 7:

“Even though the present study’s data indicated that less female participants are physically active, older married females had slightly higher the odds of involving in physical activity. We anticipated that this might be related to the fact that older married females are more physically involved in family domestic activities.”

We have also added to the discussion section (page 25) the below sentences to reflect added information on table 8:

“This might help explaining our findings which found older participants who lived outside of the country’s capital had a slightly higher the odds of eating healthy since they might be less likely to adopt Western-style diet.”

VERSION 2 – REVIEW

REVIEWER	Marcia Erazo University of Chile, Chile
REVIEW RETURNED	28-Nov-2017

GENERAL COMMENTS	Congratulations for this improved version of the article Please provide in the methods section/ statistical analysis the aim to perform the multivariate analysis:  1. To identify the variables that explain PA engagement? 2. To analyze the effect of one of these variables on PA Engagement in the presence of the others? Please provide in the methods section/ statistical analysis the criteria used to include the selected variables in the multivariate regression (Theoretical or statistical criteria), and the equation used to build the model. Table 7. A list of 4 variables are shown separately in the table, even though it is said that it corresponds to multivariable analysis. Should readers understand that the adjustment variables for "Marital Status" are "age group, gender and education"?, and for age group: marital status, gender, education, and so on?. The way the data are shown seems that you built an univariate logistic model comparing different strata within the independent variable (please see "Education of Participant" as example). Please state clearly the adjustment variables. Provide as well the crude and adjusted OR (95%CI). The same for table 8.
--

REVIEWER	Sharon Brownie Aga Khan University Kenya
-----------------	--

REVIEW RETURNED	09-Dec-2017
GENERAL COMMENTS	The manuscript is generally improved, however, grammatical errors persist, particularly in the newly inserted text, for example: Line page 9, line 35 'participants can response as follows' ... response should be respond Page 24, line 54 'participants who lived outside the capital had a slightly higher the odds of eating' requires grammatical revision Professional English editing is recommended prior to publication

REVIEWER	Lena Zimmo, PhD Aspetar, Orthopaedic and Sports Medicine Hospital Doha, Qatar
REVIEW RETURNED	10-Dec-2017

GENERAL COMMENTS	Thanks a lot.. the paper looks much better. However, the IPAQ scoring is not shown in results. Result should include the physical activity category (High, moderate, and low) as well as the MET-minutes per week. Moreover, please do the regression analysis to find out the association of health status (and more demographic characteristic's variables) with the: (1) physical activity and (2) dietary habit. P 36 of 66, L 28: How these information is related to PA and dietary habit.. you need to find the association between the demographic information/ health status with PA and dietary habit. For example: adult with high BMI reported less PA... P 38 of 66, L 35: It would be great if you also discuss the Qatar National Physical Activity Guidelines. You can also provide the Qatar Dietary Guidelines. Also, could you please add the international sedentary behavior guidelines. As you provide some data about sitting time, it would be good for the reader to know the guidelines and the current situation. P 38 of 66, L 39: what about Qatar data, please visit Stepwise report to include some data in this manuscript. P 38 of 66, L 41: When I read your sentence: "A better understanding of factors underlying health perceptions and behaviors is needed to capitalize on cardiovascular and other chronic diseases prevention efforts", I assumed you will study the physical activity behavior and the related factors, After reading the paper,. I found that you just describe the physical activity behavior. I think this is misreading sentence. P 39 of 66, L 33: you mentioned that "However, there is no specific information on the level of physical activity and dietary habits among adults in Qatar". What about Stepwise data? there is some information but it is limited and further studies are required. I think the introduction could be more organized. For example, you can focus on the health and economic consequences of physical inactivity and unhealthy dietary habit. Then you can focus on the current PA level and dietary habit nationally and internationally. Then, focus on the guidelines of PA, sedentary behavior and dietary
--

	habit. Then, you can mention the study objective. P 42 of 66, L 27: male and female is already plural. Also be careful in using male/ female or men/ women: Male and female refer to "sex"; physical and biological differences, including hormones, chromosomes and genitals, whereas men and women refer to "gender": the social roles and lifestyle played by a culture in delineating individuals as masculine or feminine. P 43 of 66, L 5: How did you measure the height, weight, and BMI? is it self-reported? P 43 of 66, L 20: The IPAQ scoring is not shown in results. Result should include the physical activity category (High, moderate, and low) as well as the MET-minutes per week. P. 45 of 66: Table 1. The characteristics of the participants: the columns male and female should be replaced with PA variables (Low moderate high) According to IPAQ. P 47 of 66, L 20: In discussion chapter, could please discuss the demographic differences between those who categorized as active and those who are inactive. Also what are the differences between those who spend 5- 10 hours setting and those who reported zero hours sitting? P 53 of 66, L 39: This table is not addressing the research question. The objective of the study was not to determine the prevalence of diseases. Health status must be studied with relation to PA and diet. I believe Table 6 (health status) can be included in Table 1; in the first column as part of the participants' characteristics). Then, find the association of these characteristics with physical activity. As I mentioned before, the columns male and female in Table 1, should be replaced with PA variables (Low moderate high) According to IPAQ. P 54 of 66, L 18: Table 7 should use IPAQ variable as outcome. It is great that you conduct the logistic regression analysis for some factors (marital status, age, gender, education, and living area) with PA or dietary habit. However, I believe you can still conduct the regression analysis to find the relation of other factors (such as nationality, level of income, and health status) with  1. Physical activity 2. Dietary habit. Given that we have very limited literature in the area of PA, I wish if you can find the relation of all of these factors with the physical activity and dietary habits, as we really need this information. P 57 of 66, L 3: In the discussion, it is important to compare the study findings (MVPA level/ dietary habit) with the guidelines and other research studies. It is also required to discuss how the reported demographic and health status are related to PA and dietary habit. You provide some data about how PA/ dietary habit are different among male and female but with the current data, you can provide more. For example, did those with high income reported better dietary habit? did they reported higher MVPA? Are people with diabetes reported higher MVPA? Are Levant more active than the Qatari? It is nice to provide some factors which could shape PA and dietary behavior but remember this is not the main objective of the study.
--	---

	P 57 of 66, L 53: Also, literature discussed the:  • Care giving responsibilities (Donnelly et al., 2011, Donnelly et al., 2012, Mabry et al., 2014). • The need for approval to engage in outdoor PA (Mabry et al., 2010, Ali et al., 2010, Mabry et al., 2014). • The Abaya (Mabry et al., 2014)
--	--

VERSION 2 – AUTHOR RESPONSE

Editor’s comments:

- Please revise the ‘Strengths and limitations’ section of your manuscript provided after your abstract. This section should relate specifically to the methods, and should not include a general summary of the study. Your first and second bullet points DO NOT relate to the methods of the study and must be removed from this section.

Response: We deleted the first and second bullet as per the Editor’s comments and edited the three points.

Reviewer(s)' Comments to Author:

Reviewer: 2

Reviewer Name: Marcia Erazo

Institution and Country: University of Chile, Chile

Please state any competing interests or state ‘None declared’: None declared

Please leave your comments for the authors below

Congratulations for this improved version of the article

Please provide in the methods section/ statistical analysis the aim to perform the multivariate analysis:

Response: On page 10, we added:

“Multivariate logistic regression analysis was performed to identify the variables that predict physical activity (PA), healthy diet (HD) engagement, and IPAQ group. Independent variables, such as living area, marital status, nationality, age group, education, health status, household income and sex were selected using the criteria for the method of forward stepwise (Wald χ^2 , $p_{in}=0.05, p_{out}=0.10$). The equation used to build the model is: $\ln(p/(1-p))=\alpha_0+\alpha_1x_1+ \alpha_2x_2+ \dots +\alpha_kx_k$, where p =probability of PA,

HD engagement and IPAQ group respectively, x_1, x_2, \dots, x_k are significant predictors after forward stepwise logistic regression procedure. Statistical significance levels were established at $\alpha=0.05$.

Please note that there are four (4) additional tables are included in this revised manuscript, Table numbers are re-numbered accordingly.

1. To identify the variables that explain PA engagement?

Response: The variables are listed on page 26.

2. To analyze the effect of one of these variables on PA Engagement in the presence of the others?

Response: Yes, we are using forward stepwise logistic regression, table 9 results show the significant predictor in the presence of the others. Adjusted ORs address this, crude OR is also reported for each significant predictor (i.e. just predictor by itself without adjusting for the presence of others in the equations), please see Table 9 (page 27 and page 28).

Please provide in the methods section/ statistical analysis the criteria used to include the selected variables in the multivariate regression (Theoretical or statistical criteria), and the equation used to build the model.

Response: On page 26, we indicated that we used forward stepwise (Wald's χ^2) for choosing predictors based on criteria of $P_{IN}=0.05$, $P_{OUT}=0.10$.

Table 7. A list of 4 variables are shown separately in the table, even though it is said that it corresponds to multivariable analysis. Should readers understand that the adjustment variables for "Marital Status" are "age group, gender and education"?, and for age group: marital status, gender, education, and so on?. The way the data are shown seems that you built an univariate logistic model comparing different strata within the independent variable (please see "Education of Participant" as example). Please state clearly the adjustment variables. Provide as well the crude and adjusted OR (95%CI). The same for table 8.

Response: Yes, sorry for the ambiguous presentations, we did mean multivariate regression analysis in our logistic regression results. Crude and Adjusted OR (95%) are reported as well. Please see Tables 9 and 10 on page 27 - page 30. (Please note that the original Tables 7 and 8 become Tables 9 and 10 respectively due to additional tables in this revised manuscript).

Reviewer: 1

Reviewer Name: Sharon Brownie

Institution and Country: Aga Khan University, Kenya

Please state any competing interests or state 'None declared': I have no competing interests

Please leave your comments for the authors below

The manuscript is generally improved, however, grammatical errors persist, particularly in the newly inserted text, for example:

Line page 9, line 35 'participants can response as follows' ... response should be respond

Page 24, line 54 'participants who lived outside the capital had a slightly higher the odds of eating' requires grammatical revision

Professional English editing is recommended prior to publication

Response: This comment is addressed throughout the whole manuscript. We have review and edited the manuscript as suggested.

Reviewer: 3

Reviewer Name: Lena Zimmo, PhD

Institution and Country: Aspetar, Orthopaedic and Sports Medicine Hospital, Doha, Qatar

Please state any competing interests or state 'None declared': None declared

Please leave your comments for the authors below

1. Thanks a lot.. the paper looks much better. However, the IPAQ scoring is not shown in results. Result should include the physical activity category (High, moderate, and low) as well as the MET-minutes per week. Moreover, please do the regression analysis to find out the association of health status (and more demographic characteristic's' variables) with the: (1) physical activity and (2) dietary habit.

Response: Descriptive statistics of Four MET-minutes/week and IPAQ grouping(High, Moderate, and Low) based on IPAQ scoring protocol are reported on page 13.

Re: regression analysis comment:

Please see: Table 9, Table 10 on page 27-30, we also add Health Status along with other demographic variables in the list of potential predictors.

2. P 36 of 66, L 28: How these information is related to PA and dietary habit.. you need to find the association between the demographic information/ health status with PA and dietary habit.

Response: On page 6. We changed the sentence to: "Given that obesity is linked to physical inactivity and/or unhealthy diet and that the fundamental cause of obesity and overweight is an energy imbalance between calories consumed and calories burned"

Please see Table 5 and discussion of Table 5 on page 20, it looks at relationships between BMI categories with PA and Dietary Habit.

3. P 38 of 66, L 35: It would be great if you also discuss the Qatar National Physical Activity Guidelines. You can also provide the Qatar Dietary Guidelines. Also, could you please add the international sedentary behavior guidelines. As you provide some data about sitting time, it would be good for the reader to know the guidelines and the current situation.

Response:

Qatar Nation Physical Activity Guidelines addressed on page 4.

Qatar dietary guidelines addressed on page 5.

Sedentary guidelines addressed on page 18.

4. P 38 of 66, L 39: what about Qatar data, please visit Stepwise report to include some data in this manuscript.

Response: Stepwise report data included on page 5 paragraph 1 (PA) and on page 5 paragraph 2 (dietary).

5. P 38 of 66, L 41: When I read your sentence: "A better understanding of factors underlying health perceptions and behaviors is needed to capitalize on cardiovascular and other chronic diseases prevention efforts", I assumed you will study the physical activity behavior and the related factors, After reading the paper,. I found that you just describe the physical activity behavior. I think this is misreading sentence.

Response: On page 6, this sentence is revised to:

"Currently, there is limited information on the level of physical activity and dietary habits among adults in Qatar. The main purpose of this study was to determine the current physical activity levels and dietary habits, and understand the variables that might predict physical activity and healthy eating behaviours among Arabic speaking adults living in the State of Qatar."

6. P 39 of 66, L 33: you mentioned that "However, there is no specific information on the level of physical activity and dietary habits among adults in Qatar". What about Stepwise data? there is some information but it is limited and further studies are required.

Response: We have addressed this with the previous comment.

7. I think the introduction could be more organized. For example, you can focus on the health and economic consequences of physical inactivity and unhealthy dietary habit. Then you can focus on the current PA level and dietary habit nationally and internationally. Then, focus on the guidelines of PA, sedentary behavior and dietary habit. Then, you can mention the study objective.

Response: We reorganized the introduction. We focused on the health and economic consequences of the unhealthy lifestyles. Then focused on PA, guidelines, and PA levels. Then focused on Diet, and available information from literature. Then mentioned the study objectives.

8. P 42 of 66, L 27: male and female is already plural. Also be careful in using male/ female or men/ women: Male and female refer to "sex"; physical and biological differences, including hormones, chromosomes and genitals, whereas men and women refer to "gender": the social roles and lifestyle played by a culture in delineating individuals as masculine or feminine.

P 43 of 66, L 5: How did you measure the height, weight, and BMI? is it self-reported?

P 43 of 66, L 20: The IPAQ scoring is not shown in results. Result should include the physical activity category (High, moderate, and low) as well as the MET-minutes per week.

Response: We revised and used male and female throughout the paper.

On page 17 we added: "The study's research assistant measured each participant height and weight and calculated the participant's BMI according to the guideline."

9. P. 45 of 66: Table 1. The characteristics of the participants: the columns male and female should be replaced with PA variables (Low moderate high) According to IPAQ.

Response:

We added a new table (which is Table 2) to address this suggestion.

Please go to Table 2 from page 13 to page 16, demographic variables, health status and medical history are included.

10. P 47 of 66, L 20: In discussion chapter, could please discuss the demographic differences between those who categorized as active and those who are inactive. Also what are the differences between those who spend 5- 10 hours setting and those who reported zero hours sitting?

Response: Regarding sitting 0 hours versus 5-10 hrs. /week-day sitting, it is a good suggestion.

Unfortunately, we only has 4 participants with 0 hours sitting/week-day, therefore it is not too meaningfully to perform such analysis.

11. P 53 of 66, L 39: This table is not addressing the research question. The objective of the study was not to determine the prevalence of diseases. Health status must be studied with relation to PA and diet. I believe Table 6 (health status) can be included in Table 1; in the first column as part of the participants' characteristics). Then, find the association of these characteristics with physical activity. As I mentioned before, the columns male and female in Table 1, should be replaced with PA variables (Low moderate high) According to IPAQ.

Response: Yes, please see one of our new tables, namely Table 2 from page 13 to Page 16.

In predicting PA & DIET, Health Status is also included along with other demographic variables, please see Tables 9 and 10 from page 27-page 30.

12. P 54 of 66, L 18: Table 7 should use IPAQ variable as outcome.

It is great that you conduct the logistic regression analysis for some factors (marital status, age, gender, education, and living area) with PA or dietary habit. However, I believe you can still conduct the regression analysis to find the relation of other factors (such as nationality, level of income, and health status) with

1. Physical activity

2. Dietary habit.

Given that we have very limited literature in the area of PA, I wish if you can find the relation of all of these factors with the physical activity and dietary habits, as we really need this information.

Response: Yes, please see Tables 9 (PA), Table 10 (Eating Habit). Table 11 (IPAQ groups), from page 27-page 31.(Please note that original Tables 7,8 becomes Tables 9 and 10 respectively due to additional tables in the revised manuscript.

13. P 57 of 66, L 3: In the discussion, it is important to compare the study findings (MVPA level/ dietary habit) with the guidelines and other research studies.

Response: On page 32, we added:

“Congruent with previous research and surveys done in Qatar and GCC countries, that were discussed in the beginning of this report, the findings of this study suggests that the Qatar adult

population's physical activity is insufficient to meet the amount recommended by the WHO and Qatar Physical Activity Guidelines.”

It is also required to discuss how the reported demographic and health status are related to PA and dietary habit. You provide some data about how PA/ dietary habit are different among male and female but with the current data, you can provide more. For example, did those with high income reported better dietary habit? did they reported higher MVPA? Are people with diabetes reported higher MVPA? Are Levant more active than the Qatari?

It is nice to provide some factors which could shape PA and dietary behavior but remember this is not the main objective of the study.

Response: Please see page 27 to page 31, we included the mentioned variables in a set of potential predictors to predict (each of PA, Eating Habits and IPAQ group). Significant ones are reported in Tables 9, 10 & 11 respectively.

14. P 57 of 66, L 53: Also, literature discussed the:

- Care giving responsibilities (Donnelly et al., 2011 [14], Donnelly et al., 2012 [15], Mabry et al., 2014)[5].
- The need for approval to engage in outdoor PA (Mabry et al., 2010 [5], Ali et al., 2010 [23], Mabry et al., 2014[5]).
- The Abaya (Mabry et al., 2014) [5]

Response: We have included all these references on page 32.

VERSION 3 – REVIEW

REVIEWER	Lena Zimmo, PhD Aspetar, Orthopaedic and Sports Medicine Hospital Doha, Qatar
REVIEW RETURNED	18-Feb-2018

GENERAL COMMENTS	Thanks a lot.. The paper looks much better. Good luck
---

REVIEWER	Marcia Erazo University of Chile, Chile
REVIEW RETURNED	26-Feb-2018

GENERAL COMMENTS	Congratulations for this version, it is very complete and gentle to read
--